# In Vitro Cultures of *Scutellaria brevibracteata* subsp. *subvelutina* as a Source of Bioactive Phenolic Metabolites

**DOI:** 10.3390/molecules28041785

**Published:** 2023-02-14

**Authors:** Inga Kwiecień, Aleksandra Łukaszyk, Natalizia Miceli, Maria Fernanda Taviano, Federica Davì, Elżbieta Kędzia, Halina Ekiert

**Affiliations:** 1Department of Pharmaceutical Botany, Faculty of Pharmacy, Jagiellonian University Medical College, 9 Medyczna St., 30-688 Kraków, Poland; 2Department of Chemical, Biological, Pharmaceutical and Environmental Sciences, University of Messina, Viale F. Stagno d’Alcontres, 31, 98166 Messina, Italy; 3Foundation “Prof. Antonio Imbesi”, University of Messina, Piazza Pugliatti 1, 98122 Messina, Italy; 4Department of Bioproducts Engineering, Institute of Natural Fibres and Medicinal Plants, National Research Institute, 71B Wojska Polskiego St., 60-630 Poznań, Poland

**Keywords:** *Scutellaria brevibracteata* subsp. *subvelutina*, in vitro culture, PGRs testing, antioxidant activity, antibacterial properties, antifungal activity, biosafety

## Abstract

Some of the more than 350 *Scutellaria* species, such as *S. baicalensis* and *S. lateriflora*, have been used in traditional medicine and today play an important role in official phytotherapy. Other species have been less investigated, and their therapeutic potential is unknown. This is one of the few studies on *Scutellaria brevibracteata* subsp. *subvelutina*, and the first research of this species’ in vitro cultures. The aim of this study was to establish an in vitro culture and analyse its phytochemical profile and biological activity. In the methanolic extracts from biomass cultured on six solid Murashige and Skoog (MS) medium variants supplemented with different combinations of 6-benzylaminopurine (BAP) and 1-naphthaleneacetic acid (NAA) in the range 0.5–3 mg/L analysed by HPLC, the presence of specific flavonoids (baicalein, baicalin, wogonin, wogonoside, scutellarin, chrysin), phenylpropanoid glycosides (verbascoside, isoverbascoside), and phenolic acids (*p*-hydroxybenzoic, caffeic, ferulic, *m*-coumaric acids) was confirmed. The dominant metabolites were wogonoside and verbascoside with the highest content of 346 and 457 mg/100 g DW, respectively. Thus, the extract with the highest content of bioactive metabolites was selected for further research and subjected to evaluation of antioxidant and antimicrobial potential. The extract exhibited good free radical scavenging activity (IC_50_ = 0.92 ± 0.01 mg/mL) and moderate reducing power and chelating activity. The brine shrimp lethality bioassay proved its lack of biotoxicity. Antimicrobial activity was tested against sixteen strains of Gram-positive and Gram-negative bacteria and fungi. The strongest growth inhibitory activity was observed against *Trichophyton tonsurans*.

## 1. Introduction

The genus *Scutellaria* L. belonging to the *Lamiaceae* family is represented by 350 species distributed worldwide [1]. Some authors give a number of 470 species, and together with lower taxa, for example, subspecies, varieties, and forms, the number equals 580 [2,3]. Some species of *Scutellaria* have been known and used in unconventional medicine for more than 2000 years. An example is *Scutellaria baicalensis* Georgii, whose root has been used in traditional Chinese medicine in the treatment of respiratory infections, hypertension, diarrhoea, haemorrhages, dysentery, and insomnia [1,4]. Currently, monographs of this raw material appear in European [5], Chinese [6], Japanese [7], and Korean [8] pharmacopoeias. This monograph can be found also in European Union countries’ national pharmacopeias. In addition, *S. baicalensis* has a monograph in the International Pharmacopoeia published by the WHO [9]. Another example is *Scutellaria barbata* D. Don. growing in the southern part of China, which is used in traditional medicine to treat snake bites, haemorrhoids, pain and swelling of the throat, and eczema, as well as inflammation and liver cancer [1]. On the other hand, *S. laterifolia* L. has been reported as traditionally used by Native Americans against neurological disorders such as epilepsy, convulsions, hysteria, and insomnia and is used nowadays in sedative herbal preparations [10,11]. Due to the close relationship between the *Scutellaria* sp., other species may also have promising high therapeutic potential. So far, 649 compounds have been identified from more than 70 species of *Scutellaria*, including 318 flavonoids, 274 terpenoids, and more than 57 other compounds, e.g., iridoids and their glycosides, phenylpropanoid glycosides, and phenolic acids [12]. The most characteristic are flavonoids of the *Scutellaria* genus such as baicalin, baicalein, wogonin, and wogonoside.

According to the newest classification [13] *Scutellaria brevibracteata* subsp. *subvelutina* (Rech.f.) Greuter & Burdet. is the accepted name of an infraspecific taxon of the species *Scutellaria brevibracteata* Stapf in the genus *Scutellaria*. The previous botanical name, the basionym of this plant, was *Scutellaria subvelutina* Rech.f. [14]. Natural sites of this subspecies are Asia Minor, in the regions of Palestine, Israel, Syria, Lebanon, Turkey, and Saudi Arabia. It prefers shady, limestone cliffs and rocky slopes at an altitude of 400–1660 m above sea level [15]. This taxon is one of four represented *Scutellaria brevibracteata* Stapf. The other subspecies are *S. brevibracteata* subsp. *brevibracteata*, *S. brevibracteata* subsp. *pannosula* (Rech.f.) Greuter & Burdet, both endemic to Turkey, and *S. brevibracteata* subsp. *icarica* (Rech.f.) Greuter & Burdet, endemic to Samos and Ikaria in the East Aegean islands [16,17].

Ethnobotanically, little is known about the traditional uses of this species. According to Turkish authors, haemostatic, tonic, and astringent properties of the leaves are used for healing the wound and as remedies against tumours and haemorrhoids [18,19]. The phytochemical composition of the plant described only a few articles [2,16,20,21,22], of which, three of them referred to essential oil and volatile compounds [2,16,22]. According to previous professional pharmacological studies, *Scutellaria brevibracteata* Stapf methanolic and aqueous extracts of aerial parts exhibit antioxidant and anti-inflammatory properties [20,23,24].

Plant biotechnology offers excellent possibilities for conducting various types of research. In the case of plants distributed in small populations or endangered species, it is doubly important. On the one hand, it allows unlimited research of a given species using biomass cultured in vitro. The usage of all available methods, from basic to very advanced, is possible [25]. In order to test the content of metabolites and then increase their production, the in vitro culturing process should be optimised by selecting the appropriate basic composition of the medium, plant growth regulators, or physical growth conditions, such as photoperiod or light intensity. Subsequently, to improve biotechnological metabolite production, different strategies could be used, such as cell line selection, cell immobilisation, precursor feeding, elicitation, biotransformation, metabolic engineering, and genetic modifications [26]. All of these studies can be carried out without endangering the natural population of the species selected for testing. Plant biotechnology is also a very good tool for the propagation and cultivation of endangered or unavailable plants, among them medicinal plants with high therapeutic value. Natural populations can be restored or supported by micropropagation [27].

The purpose of this study was to establish an in vitro culture of *Scutellaria brevibracteata* subsp. *subvelutina* and optimise its basal culturing conditions for growth and metabolite production. Moreover, the antioxidant, antibacterial, and antifungal activities of the biomass extract with the highest content of bioactive metabolites were evaluated. Finally, using the brine shrimp lethality bioassay, the biosafety of the extract was examined. To our knowledge, this is the first report on in vitro cultures of *Scutellaria brevibracteata* and at the same time one of the few studies dedicated to this species.

## 2. Results

### 2.1. Appearance and Growth of Biomass Cultured In Vitro

Cultures of *Scutellaria brevibracteata* subsp. *subvelutina* were grown on variants of MS medium containing plant growth and development regulators—PGRs (6-benzylaminopurine—BAP/1-naphthaleneacetic acid—NAA) in various concentrations: 3.0/1.0, 2.0/2.0, 1.0/1.0, 1.0/0.5, 1.0/0.0, and 0.5/2.0 (mg/L). All of them were shoot-differentiated callus cultures. The biomass was lively green with numerous branched shoots (Figure 1). The callus tissue was pale green or creamy brown in colour. In the case of cultures growing on media with the addition of 3.0 mg/L BAP + 1.0 mg/L NAA and 0.5 mg/L BAP + 2.0 mg/L NAA, the shoots were shorter, while the appearance of biomasses growing under the remaining culture conditions was comparable. Their shoots were longer and branched.

During 4-week culture cycles on the tested medium variants, in vitro biomasses showed a not very varied increase (Table 1). The highest increases in fresh and dry biomasses were found on the medium supplemented only with 1.0 mg/L of BAP at 4.1-fold and 4.36-fold, respectively. A comparable result in dry biomass was obtained on the medium variant with 2.0 mg/L BAP + 2.0 mg/L NAA. It was 4.23-timefold. The least favourable for biomass growth were the medium variants containing 0.5 mg/L of BAP + 2.0 mg/L of NAA and 1.0 mg/L of BAP + 0.5 mg/L of NAA, with 3.93- and 3.91-fold increment of dry biomass, respectively. However, the differences in the results obtained were not statistically significant.

### 2.2. Endogenous Accumulation of Metabolites

*Scutellaria brevibracteata* subsp. *subvelutina* methanolic extracts from in vitro cultures showed the presence of metabolites from the flavonoid, phenylpropanoid glycoside, and phenolic acid groups. These were baicalein, baicalin, wogonin, wogonoside, scutellarin, and chrysin (specific for *Scutellaria* genus flavonoids); *p*-hydroxybenzoic, caffeic, ferulic, and *m*-coumaric acid (phenolic acids); and verbascoside and isoverbascoside belonging to phenylpropanoid glycosides.

The total flavonoid content ranged from 384.6 to 574.8 mg/100 g of DW (Table 2). The highest content was obtained in the methanolic extracts of *S. b.* subsp. *subvelutina* in vitro cultures grown on medium variant containing 3.0 mg/L BAP and 1.0 mg/L NAA. It was 1.49 times higher than in the biomass grown on the medium supplemented with 1.0 mg/L of BAP and 1.0 mg/L of NAA (the lowest content of flavonoids). The dominant metabolite in all biomass extracts was wogonoside. The content of this compound ranged from 195 to 346.4 mg/100 g of DW. The baicalin content was also high and ranged from 66.2 to 104.5 mg/100 g of DW. Wogonoside and baicalin, both with glucoside structure, are glucuronide derivatives of wogonin and baicalein, respectively. The content of the other flavonoids—wogonin, scutellarin, and chrysin—reached the range of 10–40 mg/100 g DW. Among the detected flavonoids, the baicalein content was the lowest from 10 to 15 mg/100 g DW.

Analysis of the phenolic acid content showed a low amount of these compounds in the extracts. The total phenolic acid content in biomass ranged from 9.8 to 15.5 mg/100 g of DW (Table 3). The highest amount was confirmed in the extracts of biomass grown on medium supplemented with 2.0 mg/L of BAP and NAA each. The dominant compound in all extracts was ferulic acid (5.1–10.1 mg/100 g DW). Caffeic and *p*-hydroxybenxoic acid contents were confirmed in the low amount that did not exceed 2.6 mg/100 g DW. *m*-Coumaric acid was detected only in two extracts, from biomasses grown on the medium containing 2.0 mg/L BAP + 2.0 mg/L NAA and 0.5 mg/L BAP + 2.0 mg/L NAA (2.76 and 2.9 mg/100 g DW, respectively).

Data on phenylpropanoid glycoside content showed that it was one of the main metabolite groups in the extracts analysed (Table 4). The maximum total content of verbascoside and isoverbascoside ranged from 365 mg/100 g DW (medium variant with 0.5 mg/L BAP + 2.0 mg/L NAA) to 457.4 mg/100 g DW (medium variant with 3.0 mg/L BAP + 1.0 mg/L NAA). Two phenylpropanoid compounds were detected in all extracts. In biomass grown on five medium variants, the verbascoside content was almost the same and ranged from 350 to 377 mg/100 g DW. Only in biomass from medium supplemented with 3.0 mg/L BAP and 1.0 mg/L NAA was the amount of verbascoside higher (440.6 mg/100 g DW). The isoverbascoside content was significantly lower than the verbascoside. It was in the range of 15.3 to 19.8 mg/100 g of DW.

In summary, the medium variant containing 3.0 mg/L of BAP and 1.0 mg/L of NAA was the most conducive to the accumulation of the metabolites analysed, especially flavonoids and phenylpropanoid glycosides, in the biomass of *S. b.* subsp. *subvelutina* in vitro cultures.

### 2.3. Activity of the Selected Extract

An extract with the highest content of the tested metabolites was selected for further research dedicated to biological activity. Extract of in vitro shoot cultures of S. *brevibracteata* subsp. *subvelutina* grown on MS medium variant A (3.0 mg/L BAP and 1.0 mg/L NAA) was used to test its antioxidant, antibacterial, and antifungal activity. Additionally, the biotoxicity of the extract was evaluated.

#### 2.3.1. Antioxidant Activity and Total Phenolic Content

The antioxidant properties of the *S. brevibracteata* subsp. *subvelutina* methanolic extract were assessed using three different tests (DPPH test; reducing power test; Fe^2+^ chelating activity) to determine the possible antioxidant mechanisms of action of the phytocomplex. The results of the DPPH test, utilised to establish the free-radical-scavenging properties of the extract, are shown in Figure 2. The extract showed good dose-dependent radical scavenging activity. The activity of the extract was lower than that of BHT used as reference standard as demonstrated by IC_50_ values (IC_50_ = 0.92 ± 0.01 mg/mL and 0.07 ± 0.01 mg/mL, respectively); furthermore, the efficacy of the extract was close to that of the standard at the highest concentration tested of 2 mg/mL (91%).

The reducing power reflects the ability to stop the radical chain reaction. In this assay, the presence of antioxidant compounds in the sample determined the reduction of Fe^3+^ to the ferrous form (Fe^2+^); this reduction is highlighted by spectrophotometric measurement (700 nm) of the change of yellow colour of the test solution to various shades of green and blue, depending on the reducing power of the antioxidant sample. Figure 3 shows the results of the reducing power of methanol extract obtained from biomass of in vitro culture of *Scutellaria brevibracteata* subsp. *subvelutina*; the extract displayed moderate activity, which was dose dependent, compared to the standard BHT. The effect was lower than BHT, as also evidenced by ASE/mL values (10.15 ± 0.67 and 0.89 ± 0.06 ASE/mL, respectively) (Figure 3).

The method of Fe^2+^-chelating activity utilised the reagent ferrozine, which can quantitatively form complexes with Fe^2+^; in the presence of chelating agents, the complex formation is inhibited, with the result that the red colour of the complex is decreased. Measurement of colour reduction, therefore, allows for the estimation of the chelating activity of the coexisting chelator. In the Fe^2+^ chelating activity assay, the extract showed a low chelating property reaching approximately 30% at the highest concentration tested (Figure 4). The IC_50_ values confirmed the lower activity of the extract with respect to the standard EDTA (3.33 ± 0.05 mg/mL and 0.007 ± 0.001 mg/mL, respectively).

The results of the Folin–Ciocâlteu assay showed that the total phenolic content of the extract was 46.40 ± 0.4 mg gallic acid equivalent (GAE)/g extract.

#### 2.3.2. Antibacterial Activity

The activity of the extract analysed against Gram-positive and Gram-negative bacterial strains was not very diverse (Table 5). The highest antibacterial activity of the extract was confirmed against the Gram-positive rod *Bacillus subtilis* (MIC = 10 mg/mL). For the other strains, the minimal inhibitory concentration was 30 mg/mL. Only for *Staphylococcus aureus* was the effect slightly stronger (MIC = 20 mg/mL). In the group of Gram-negative rods, the MIC values determined were 30 mg/L. Only for *Pseudomonas aeruginosa* was the obtained result slightly better. The inhibitory concentration of the tested extract was equal to 20 mg/L. It can be considered that the extract from in vitro cultures of *S. brevibracteata* subsp. *subvelutina* has moderate antibacterial activity.

#### 2.3.3. Antifungal Activity

The research showed that the in vitro culture extract indicated the same antifungal activity on the three species of *Candida* (MIC = 30 mg/mL). Similar results were obtained when testing the effect on mould fungi. The MIC value obtained against *Aspergillus flavus* was 30 mg/mL, while against *Penicillium chrysogenum* was 40 mg/mL. The strongest antifungal activity of the tested extract was shown against the dermatophyte. The growth of *Trichophyton tonsurans* was inhibited at a lower extract concentration of 7.5 mg/mL (Table 6).

#### 2.3.4. Artemia Salina Lethality Bioassay

The brine shrimp larva (*Artemia salina*) is an invertebrate used in the lethality bioassay, considered a useful tool for preliminary assessment of toxicity. This bioassay has the advantage of being rapid, inexpensive, and simple [28]. It represents a simple technique for predicting the toxicity of plant extracts in order to consider their safety. The results of the assay showed that *S. brevibracteata* subsp. *subvelutina* biomass extract did not display any toxicity against brine shrimps (Appendix A). Indeed, the median lethal concentration values were found to be above 1000 µg/mL, thus indicating their potential safety according to Clarkson’s toxicity criterion [29].

## 3. Discussion

### 3.1. In Vitro Cultures

This is the first report of *S. brevibracteata* subsp. *subvelutina* in vitro cultures. The dry biomass growth that we obtained from the solid cultures after 4-week growing cycles (3.91–4.36-timefold) was satisfactory compared to the biomass increases for other *Scutellaria* solid cultures maintained in our laboratory, *S. baicalensis* at 4.47–7.89-fold [30] and *S. lateriflora* at 2.22–3.69-fold [31]. For larger growth, further optimisation of another culture type, e.g., agitated or in bioreactors, is needed.

The richness and diversity of flavonoid compounds in plant species of the genus *Scutellaria* are very large, but only a few of them occur in concentrations high enough to be responsible for biological activity. Despite the remarkable similarities in the qualitative composition of the secondary metabolites, most of all in flavonoids, of the most known *S. lateriflora* and *S. baicalensis*, there are differences between these species in the quantitative containment of these metabolites [32]. On the basis of numerous literature studies, it should be assumed that the little-known and unstudied *Scutellaria* species are also characterised by a similar qualitative metabolite composition. Our obtained results with *S. b.* subsp. *subvelutina* confirmed this hypothesis. The main metabolites were specific *Scutellaria* flavonoids.

Apart from the articles on the composition of essential oil and volatile compounds, only three studies have reported on the remaining secondary metabolites that occur in *S. brevibracteata*. Franzyk et al. [21] focused in 1997 on iridoid compounds. In the water-soluble fraction of the ethanol extract, the presence of scutelliaroside I, the catalpol *t*-cinnamoyl ester, was confirmed. The newer work of Erdogan confirmed in the methanolic extract of *Scutellaria brevibracteata* subsp. *subvelutina* aerial parts the presence of many iridoid glycosides, including catalpol, globularin, scutellarioside II, picroside III, pensteminoside, 8-epiloganic acid, albidoside, and agnucastoside B. In addition to them, they found martynoside belonging to phenylpropanoid glycosides and flavonoids such as hispidulin, its 7-O-glucuronide, and luteolin 7-O-glucuronide [20]. O-glycosylating by glucuronic acid at position 7 flavonoids are typical for the *Scutellaria* genus. The article by Dogan et al. focused on the aqueous extract of *Scutellaria brevibracteata* obtained from aerial parts. The authors detected metabolites belong to iridoid derivatives: loganic acid, 6′-O-*E*-caffeoyl mussenosidic acid, 6′-O-*E*-*p*-coumaroyl gardoside, albidoside, and globularin (scutellarioside I). They also found flavonoids, such as luteolin, luteolin 7-O-β-glucuronide, apigenin, 3-O-methyl kaempferol, trihydroxy-dimethoxyflavone, and oroxylin A [23].

The aerial parts and roots of the *Scutellaria* species are flavonoid raw materials. More than 150 metabolites from this group have been identified in various species, mainly with the structure of flavones. The main flavonoid aglycones found in this genus include baicalein (5,6,7-trihydroxyflavone), wogonin (5,7-dihydroxy-8-methoxyflavone), scutellarein (5,6,7,4′-tetrahydroxyflavone), chrysin (5,7-dihydroxyflavone), and oroxylin A (5,7-dihydroxy-6-methoxyflavone). They are accompanied by their glycosidic derivatives [33]. In addition to flavones, the presence of flavonols, flavanones, flavanonoles, chalcones, flavonolignans, and biflavonoids has been confirmed in representatives of this genus [1,34]. In the current research, we confirmed the presence of typical flavonoids of *Scutellaria* in the analysed extracts, such as baicalein, baicalin, wogonin, wogonoside, scutellarin, and chrysin. Quantitative comparison with the same type of culture of other species of *Scutellaria* studied by us shows that the total flavonoid content in *S. brevibracteata* subsp. *subvelutina* (574.8 mg/100 g of DW) is lower than in *S. lateriflora* (722.0 mg/100 g of DW) [31] but higher than in *S. baicalensis* solid cultures (428.6 mg/100 g DW) [30]. The best results were obtained on different variants of MS medium containing 3.0 mg/L BAP + 1.0 mg/L NAA for *S. brevibracteata* subsp. *subvelutina*, 1.0 mg/L BAP + 0.5 mg/L NAA for *S. lateriflora*, and 0.5 mg/L BAP + 2.0 mg/L NAA for *S. baicalensis*. This is the typical situation in biotechnological studies. The best production media always should be selected experimentally for the in vitro culture of each species.

Phenolic acids in the plant material of *Scutellaria* species were detected only a few times. These were caffeic, chlorogenic, *p*-coumaric, ferulic, *p*-hydroxybenzoic, protocatechuic, vanillic, and rosmarinic acids, and additionally the biogenetic precursor of one subgroup of phenolic acid—cinnamic acid [1,34,35,36]. Now, we confirm for the first time the presence of four phenolic acids (*p*-hydroxybenzoic, caffeic, ferulic, and *m*-coumaric) in the extracts of *S. brevibracteata* subsp. *subvelutina*. In our previous articles, we showed only the presence of one compound—3,4-dihydroxyphenylacetic acid—in *S. lateriflora* and *S. baicalensis* in vitro cultures but grown on a different medium (acc. Linsmaier and Skoog). On MS medium variants, these in vitro cultures produced any phenolic acids [30,31].

The phenylpropanoid glycosides identified in various species of *Scutellaria* are verbascoside, leucosceptoside A, martynoside, isomartynoside, salidroside, and darenoside A and B. Some older reports stated that verbascoside has not occurred in this taxon, and its presence is the result of material identification errors [1]. In all our previous studies with *S. baicalensis* and *S. lateriflora*, we confirmed the presence of verbascoside and the lack of isoverbascoside in in vitro culture extracts. The best production media were also different for each in vitro culture. The maximum content of this compound was 381.7 mg/100 g of DW for *S. lateriflora* (MS, 1.0 mg/L BAP + 1.0 mg/L), 830.9 mg/100 g of DW for *S. baicalensis* (MS, 1.0 mg/L BAP + 0.5 mg/L NAA), and 440.6 mg/100 g of DW for *S. brevibracteata* subsp. *subvelutina* (MS, 3.0 mg/L BAP + 1.0 mg/L NAA). In the extract of *S. brevibracteata* subsp. *subvelutina*, we confirmed the presence of isoverbascoside as well. The content of this metabolite was 18 to 26 times lower than the content of verbascoside.

Shoot cultures grow slowly, and such cultures can accumulate more secondary metabolites compared to fast-growing callus and cell cultures. Organogenesis, the development of shoot buds from unorganised callus, is a complex phenomenon controlled by cytokinins (in combination with auxins). Thus, it affects genes related to cell, tissue, and meristem differentiation. PGRs not only control fundamental growth and developmental processes but are also known to regulate the production of plant secondary metabolites in plant tissue culture. Numerous examples of this phenomenon have been reported in the literature [37]. Cytokinins are perceived by membrane-localised histidine-kinase receptors and transduced through a His-Asp phosphorelay to activate a family of transcription factors in the nucleus. The transcriptional response to exogenous cytokinin results in substantial gene expression changes shown in rice and *Arabidopsis* [38]. Exogenous application of high concentrations (2.25–22.5 mg/L) of BAP to *Arabidopsis* plants before inoculation with pathogens led to decreased susceptibility to infection [39]. Transcriptional factors play a role in plant defence control by detecting stress signals. The accumulation of secondary metabolites is one of the responses [40]. The main biosynthesis route starts from the shikimate pathway, which is the initial pathway for the biosynthesis of aromatic amino acids (phenylalanine, tyrosine, and tryptophan). Among them, phenylalanine is the main one, which is the substrate for the biosynthesis of phenolic secondary metabolites. The key enzyme of this pathway is phenylalanine ammonia-lyase [41]. Chalcone synthase and chalcone isomerase are the main enzymes involved in the further steps of flavonoid biosynthesis. A different class of these enzymes was discovered in the shoots and roots of *S. baicalensis* [42]. The expression of genes related to the biosynthetic pathway of phenolic metabolites was detected under cytokinin impact in *Vitis vinifera* [43]. The whole process of secondary metabolite biosynthesis induced by PGRs is obviously multistage and complex. It may be different for each plant species, and in in vitro cultures, it may also be different.

The differences in metabolites content between our results and the results mentioned above of other authors who studied *Scutellaria brevibracteata* grown in open air may be due to the fact that (1) we analysed the crude, unfractionated methanolic extract; (2) our analytical method was focused on phenolic and polyphenolic compounds such as flavonoids, catechins, or phenylpropanoid glycosides; (3) we analysed biomass in vitro cultures that could change cell metabolism under growing conditions. Taking into account all the data from the literature, a more thorough analysis of the extract is needed to identify other unknown metabolites. We plan to use the method for detecting iridoid compounds and/or spectral analyses.

### 3.2. Biological Activity of the Extract

Published data on the biological activity of *Scutellaria brevibracteata* focus on the anti-inflammatory activity of its metabolites or different fractions of the extract. Erdogan et al. tested the in vitro anti-inflammatory activity of isolated metabolites (nitrite oxide and IL-6 level) in LPS-induced RAW 264.7 macrophage cells [20]. Other authors also demonstrated the anti-inflammatory activity of *S. brevibracteata* in carrageenan-induced paw oedema test in mice [23]. Senol et al. showed that methanol extract of *S. b.* subsp. *subvelutina* inhibits tyrosinase the strongest among all 33 species tested. The extract also exhibited weak inhibition against acetylcholinesterase and butyrylcholinesterase and possessed some antioxidant activity [24].

The impairment of antioxidant protection is an important factor underlying the development of many diseases. Antioxidant activity occurs through different mechanisms; thus, the antioxidant properties of plant extract or pure compound should be evaluated by the use of various methods in order to acquire a more complete antioxidant profile. In these assays, extracts are generally assessed for their function as reducing agents, hydrogen donors, singlet oxygen quenchers, or metal chelators [44]. They are classified as primary (chain breaking) antioxidants when they react with free radicals by producing less reactive species or by interrupting the radical chain reaction; on the contrary, when they act by inhibiting the radical formation protecting against oxidative damage, they are defined as secondary (preventive) antioxidants [45]. Thus, no single testing method is capable of providing a comprehensive view of the antioxidant profile of an extract; therefore, in order to evaluate the antioxidant capacity of the extract obtained from in vitro cultured biomass of *S. brevibracteata* subsp. *subvelutina* extract, three in vitro systems based on different mechanisms of determination of antioxidant capacity were used. The primary antioxidant properties were examined using the DPPH assay, on the basis of the hydrogen atom transfer (HAT) and electron transfer (ET) mechanisms, and the reducing power, an ET-based assay. The secondary antioxidant ability was determined by measuring the ferrous ion (Fe^2+^)-chelating activity.

The results obtained by us show that the methanolic extract from in vitro cultures of *Scutellaria brevibracteata* subsp. *subvelutina* exhibits good primary antioxidant properties, being active in the DPPH test and reducing power assay. Comparing these results with those obtained for *S. baicalensis* (IC_50_ = 1.77 mg/mL and 14.0 ASE/mL, respectively; not published) and *S. lateriflora* (IC_50_ = 1.64 mg/mL and 43.48 ASE/mL, respectively) [41] in vitro culture extracts, it is evident that *S. brevibracteata* subsp. *subvelutina* extract showed better both radical scavenging activity and reducing power activity (IC_50_ = 0.92 mg/mL and 10.15 ASE/mL, respectively). However, the chelating activity was definitely the weakest (IC_50_ = 3.33 mg/mL) compared to *S. baicalensis* (IC_50_ = 0.62 mg/mL; not published) and *S. lateriflora* (IC_50_ = 0.61 mg/mL). It had been reported that antioxidant activity can be associated with the content of phenolic compounds, especially flavonoids, in plant material [46,47]. Therefore, in this study, the total phenolic content of *S. brevibracteata* subsp. *subvelutina* extract was evaluated. The total phenolic content of this extract was 46.40, of *S. baicalensis* was 50.24, and for *S. lateriflora* was 37.27 mg (GAE)/g of extract [41]. These data clearly indicate significant differences between the in vitro cultures of the three species in the mechanisms of exhibited antioxidant activity. At the same time, they indicate how important it is to perform different antioxidant tests. Senol et al. screened the methanol extracts obtained from the aerial parts of 33 *Scutellaria* taxa growing wild in Turkey. Among them, *S. brevibracteata* subsp. *subvelutina* extract showed a good total phenolic content equal to 148.89 mg GAE/g of extract, as well as high DDPH radical scavenging properties (reaching 50% DPPH radical scavenging activity at the concentration of 0.5 mg/mL) and moderate reducing power (an absorbance of 0.335 was achieved at the concentration of 1 mg/mL), as well as no activity in the ferrous ion-chelating assay [24]. Comparing these results with ours, it is evident that the extract obtained by biotechnological means, despite having a lower total phenol content, has a greater both reducing power (with absorbance values equal to 0.601 at the concentration of 1 mg/mL) and chelating activity compared to the one examined by Senol et al. [24].

The antioxidant activity of plant extracts and their biological activities are often explained by their phytochemical profile. The instrumental analysis of the extract revealed the presence of compounds belonging to different chemical classes, i.e., flavonoids and phenylpropanoids. Skullcap flavonoids have a proven, strong antioxidant effect through the ability to scavenge free radicals and the ability to chelate metal ions. These compounds also reduce the intensity of lipid peroxidation in vitro and in vivo. Many species of the genus *Scutellaria*, e.g., *S. baicalensis*, *S. altissima*, *S. alpina*, *S. orientalis*, and *S. salviifolia*, as well as isolated flavonoids from them, have been investigated for their antioxidant potential in recent years [12]. Baicalein and baicalin have the ability to scavenge free radicals, inhibit xanthine oxidase, and protect the liposome membrane from lipid peroxidation induced by light and H_2_O_2_. It was shown using different tests that only baicalein among the four main flavones of *Scutellaria* exerted a consistent antioxidant effect [48]. Wogonin has a weak inhibitory effect on xanthine oxidase. The antioxidant activity mechanisms of scutellarin are also known. This flavonoid reduces cytotoxicity and the level of lipid peroxidation in vitro by capturing superoxide free radicals [33,49]. In vivo studies in rats have shown that baicalin, baicalein, and wogonin reduce lipid peroxidation in animals treated intraperitoneally with peroxidants [1]. However, baicalin is considered to be the main responsible for the anti-inflammatory effect of the Baikal skullcap [50].

Other important compounds detected by us in the extract of *S. b.* subsp. *subvelutina* were phenylpropanoid glycosides, verbascoside and isoberbascoside, which because of their chemical structure have a strong antioxidant effect [51]. Along with that, they also possess anti-inflammatory, antibacterial, antiviral, and anticancer properties. The antioxidant activity of phenylpropanoid glycosides is related to the ability to scavenge free oxygen radicals, and thus they inhibit lipid peroxidation and nitric oxide synthesis. The presence of the *o*-dihydroxyphenyl group in their molecule has been shown to have a strong positive effect on antioxidant potential [52,53]. Several authors have demonstrated that these compounds showed great DPPH• scavenging ability as well as reducing power [54,55,56,57].

It can be hypothesised that good primary antioxidant properties observed for the ex-tract are related mainly to the phenylpropanoid verbascoside and isoverbascoside but also to the flavonoid bacailein, whose anti-radical properties have been demonstrated [58]. In vitro methods to investigate antioxidant activity are nonspecific and incomplete, and this is because complex tests are needed to demonstrate the activity of the extract and understand its biological action mechanism. Such complex tests should support the phytochemical analysis of the plant material. They are important because the biological activity of the extract is not always the sum of the potential activities of individual metabolites.

The methanolic extract of biomass from in vitro cultures of *Scutellaria brevibracteata* subsp. *subvelutina* that we tested showed moderate antibacterial activity against Gram-positive and Gram-negative strains. The extract exhibited average activity with MIC = 30 mg/mL and the strongest activity against *Bacillus subtilis* (MIC = 10 mg/mL). In the case of antifungal activity, the strongest effect was observed against *Trichophyton tonsurans* (MIC = 7.5 mg/mL). Growth inhibition of other tested yeast-like fungi and moulds was moderate.

Many compounds and extracts of *Scutellaria* species show antibacterial activity in vitro, suggesting that the entire *Scutellaria* genus has good potential as an antimicrobial medicinal raw material. *Scutellariae baicalensis radix* preparations are used due to their anti-inflammatory and antimicrobial activity in inflammatory infections of the skin and oral mucosa. The test of Baikadent (gel with *S. baicalensis* root extract, Herbapol, Wrocław) showed the sensitivity of yeast-like fungi on it. The MIC value for most of them was between 12.5 and 25.0 mg/mL. The most susceptible strains were *Candida tropicalis* and *Candida parapsilosis*, while the least sensitive was *Candida guilliermondii* (MIC ≥ 300.0 mg/mL) [59]. The same research team showed the inhibitory activity of Baicadent against Gram-positive rods isolated from atherosclerotic plaques [60].

The water extract of *S. baicalensis* had antibacterial activities that inhibited *Staphylococcus aureus* with a MIC value of 7.5 mg/mL [61]. In other research, aqueous extracts of *S. baicalensis* root have antimycotic properties against *Aspergillus fumigatus, Candida albicans*, *Geotrichum candidum*, and *Rhodotorula rubra* [62]. Although 70% ethanol extract of *S. baicalensis* was effective against *Bacillus subtilis* and *Staphylococcus aureus*, *Aspergillus niger*, and *Candida albicans* with MICs from 0.03 to 0.4 mg/mL [63]. A hot aqueous extract of *S. baicalensis* roots strongly inhibited the growth of *Alcaligenes calcoaceticus, Klebsiella pneumoniae*, *Pseudomonas aeruginosa*, and *Staphylococcus aureus* at concentrations of 0.2–0.4 mg/mL but was not active against *Escherichia coli* at concentrations of up to 1.6 mg/mL [64]. A hot aqueous extract of roots, 0.25–1.0 μg/mL, also inhibited the growth of *Actinomyces naeslundii*, *A. odontolyticus*, *Actinobacillus actinomycetemcomitans*, *Fusobacterium nucleatum*, *Bacteroides gingivalis, B. melaninogenicus*, and *Streptococcus sanguis* [65]. The extract of other *Scutellaria* species also shows antibacterial activity. The methanolic extract of aerial parts of *S. lindbergii* showed a moderate antimicrobial effect against *Staphylococcus aureus* with a MIC value of 6.25 mg/mL [66]. However, *Scutellaria barbata* herb demonstrated activity against *Acinetobacter baumannii* [67].

The essential oil and its constituents from *S. grossa* [68], *S. repens* [69], *S. edelbergii* [70], *S. barbata* [71], *S. strigillosa* [72], *S. rupestris*, and *S. sieberi* [73] reveal strong bactericidal activity. Nan et al. discussed the antimicrobial activity of the total fraction of flavones from the root extract of *S. baicalensis* [74]. Baicalein inhibited the growth of *Fusarium oxysporum* and *Candida albicans* in vitro, with MICs of 0.112 mg/mL and 0.264 mg/L, respectively [9]. Another flavone, isolated from *S. oblonga* leaves, techtochrysin, was effective against *Enterococcus faecalis*, *Shigella dysenteriae*, and *Pseudomonas aeruginosa,* with the lowest MIC of 24 μg/mL. This metabolite was also active against *Bacillus subtilis*, *Salmonella enterica*, *Vibrio cholerae*, and *Klebsiella pneumoniae* with an MIC of 32 μg/mL [75]. A more well-known flavonoid, quercitin-3-glucoside, also showed great antimicrobial activity [75]. Some species of *Scutellaria* also contain other bioactive compounds, such as alkaloids, which break down the stability of peptidoglycans and degrade the walls of fungal cells [72]. A neo-clerodane diterpenoid named scutalpin A could be significantly effective against *Staphylococcus aureus* (MIC 25 μg/mL) [76]. We have not found information in the literature on the activity against dermatophyte species.

All these data supported our results. This indicates the great potential of *Scutellaria* species to fight microorganisms. In the literature, many mechanisms of antibacterial activity of flavonoids and other phenolic compounds have been proposed [77,78]. One of them suggests a link between the lipophilicity of the compounds and their ability to cross cytoplasmic membranes [79]. According to Yuan et al., the antimicrobial activity of flavonoids results from their ability to disrupt the permeability barrier of microbial membrane structures. It can involve damage to phospholipid bilayers, inhibition of the respiratory chain, or ATP synthesis. This effect is more pronounced in Gram-positive bacteria due to the lack of an outer membrane. [79]. Complex extracts and individual compounds could be useful to support prophylactic of anticandidal therapy in the treatment of bacterial inflammations of the oral mucosa, periodontium, and teeth. They could be helpful in dismissing resistant strains to antibiotic therapy. After stimulating in vitro cultures of *Scutellaria brevibracteata* subsp. *subvelutina* to a high production of bioactive metabolites and optimising this process, which is a standard biotechnological procedure, an increase in antimicrobial activity should be expected.

Suggesting other, less-known species of *Scutellaria* for therapeutic use requires each time demonstrating a lack of toxicity of the species. It is not enough to invoke phylogenetic similarity in the chemical composition of species of this taxon. The root of *S. baicalensis* and its main flavone constituents are the most studied [80]. These phytochemicals are not only cytostatic, but also cytotoxic to various human tumour cell lines [81,82], e.g., several human prostate cancer cell lines [83], oestrogen-receptor-positive human breast cancer [84], lung cancer, liver cancer, or myeloma [33]. Most importantly, they show almost no or minimal toxicity in studies on normal cells, e.g., epithelial, myeloid, or blood cells [85]. Due to the potential anticancer activity, the biosafety of the extract is very important. Our research indicates that the methanolic extract of in vitro cultured *Scutellaria brevibracteata* subsp. *subvelutina* does not exhibit toxicity to the invertebrate species *Artemia salina* in the concentration tested (up to 1 mg/mL). Toxicological studies on animal models of *S. baicalensis* root show that the oral lethal dose (LD_50_) in mice was >2.0 g/kg (70% methanol extract) [9]. Subcutaneous LD_50_ in mice was 6.0 g/kg (ethanol extract), 6.0 g/kg for baicalin, and 4.0 g/kg for wogonin. However, the intraperitoneal LD_50_ of baicalin was 3.1 g/kg [86]. Intravenous administration of an aqueous extract in a dose of 1.0 g/kg or less does not show a lethal effect. Moreover, oral administration of an aqueous root extract dogs of 4.0 to 5.0 g/kg BW of the same extract did not cause any toxic effects [9]. It should also be remembered that *S. baicalensis* is a pharmacopeial species, and its root is approved in official European medicine [4]. There is no obvious adverse reaction in the oral administration of *Scutellaria baicalensis* preparation in humans [87]. The biosafety of other species should also be examined.

Summing up, the extract of *S. b*. subsp. *subvelutina* from the biomass of in vitro cultures is a good source of bioactive compounds with a polyphenolic structure that possesses a very valuable antioxidant and antimicrobial activity. The biomass of the in vitro culture could be proposed as a pharmaceutical raw material obtained with the use of biotechnological methods. The next step of research in this area should be to scale up and establish a bioreactor culture.

## 4. Materials and Methods

### 4.1. Establishing of In Vitro Cultures

In vitro shoot cultures of *Scutellaria brevibracteata* subsp. *subvelutina* (Rech.f.) Greuter & Burdet were derived from seeds acquired in 2018 from The Jerusalem Botanical Gardens, The Hebrew University Givat Ram Campus (voucher number 100099). The seeds were surface-sterilised using 0.1% HgCl_2_ for different time intervals (5–20 min). After sterilisation, the seeds were transferred to Murashige and Skoog (MS) [88] initiating medium containing 3% sucrose (*w*/*v*), 0.7% phytoagar (Duchefa Biochemie, the Netherlands) as a gelling agent and supplemented with 1 mg/L BAP (6-benzylaminopurine) and 0.5 mg/L NAA (1-naphthaleneacetic acid). The medium was adjusted to pH 5.7 prior to autoclaving. Shoot cultures derived from germinated seedlings were grown on the same MS medium variant under constant LED light with an intensity of 16 μmol/m^2^·s at 25 ± 2 °C and were subcultured every 4 weeks.

### 4.2. Experimental In Vitro Cultures

Experimental cultures of *S. brevibracteata* subsp. *subvelutina* (inoculum 1.2 g) were grown in 200 mL jars containing 25 mL of MS medium solidified with agar and under the same culture conditions as the initial cultures. Six medium variants containing BAP and NAA in various concentrations (Table 7) were tested. The concentrations of PGR concentrations were chosen on the basis of our previous experiment. The growing cycle was 4 weeks. For each medium variant, 9 samples were made.

### 4.3. Sample Preparation

The fresh biomass was collected and dried at room temperature. The pulverised biomass samples were extracted with methanol (50 mL) for 2 h under a reflux condenser at 78 °C. The extracts were transferred to crystallisers through filter paper (Whatman paper) and left at room temperature to evaporate the solvent.

### 4.4. RP-HPLC Analysis of Secondary Metabolites

The dry extracts were dissolved in spectral-grade methanol, filtered, and analysed by HPLC. An HPLC system (Merck–Hitachi) and a Purospher RP-18e analytical column (4 × 250 mm, 5 μm; Merck) were used. The samples were analysed using a modified HPLC method [89] described previously [90]. Briefly, the mobile phase consisted of methanol (I) and 0.5% acetic acid (II); the flow rate was 1 mL/min, and the analysis time was 90 min.

Qualification and quantification analysis were based on a comparison with 61 reference substances. Flavonoids (43): apigenin, chrysin, cynaroside, isorhamnetin, hyperoside, luteolin, myricetin, narigenin, populnin, quercetin, quercetin 7-glucoside, quercitrin, rhamnetin, robinin, rutoside, scutellarein, vitexin, wogonoside (Sigma-Aldrich^®^, St Louis, MO, USA); apigenin 5-glucoside, apigenin 7-glucuronide, apigenin 4′-rhamnoside, astragalin, avicularin, baicalin, baicalein, diosmetin, isoquercetin, kaempferol, kaempferol 3-glucorhamninoside, kaempferol 3-rhamnoside, kaempferol 7-rhamnoside, kaempferol 4′-glucoside, miquelianin, narirutin, scutellarin, wogonin (ChromaDex, Irvine, CA, USA); apigetrin, isovitexin, gujaverin, oroxylin A, sculcapflavone II, trifolin, vicenin II (ChemFaces, Wuhan, China). Phenolic acids (26): 3,4-dihydroxyphenylacetic acid, 3-hydroxyphenylacetic acid, caftaric acid, caffeic acid, chlorogenic acid, cryptochlorogenic acid, 2-coumaric, 3-coumaric, 4-coumaric acids, dihydrocaffeic acid, ellagic acid, ferulic acid, 4-O-feruloyl-quinic acid, gallic acid, gentisic acid, hydrocaffeic acid, 4-hydroxybenzoic acid, isochlorogenic acid, isoferulic acid, neochlorogenic acid, protocatechuic acid, rosmarinic acid, salicylic acid, sinapic acid, syringic acid, and vanillic acid (Sigma-Aldrich^®^, St. Louis, MO, USA). Phenylpropanoid glycosides (2): verbascoside and isoverbascoside (ChromaDex, Irvine, CA, USA).

### 4.5. Determination of Total Phenolic Content

The total phenolic content of *S. brevibracteata* subsp. *subvelutina* extract was measured by the Folin–Ciocâlteu reagent [91]. Each sample solution (100 μL) was mixed with 0.2 mL Folin–Ciocâlteu reagent, 2 mL of distilled water, and 1 mL of 15% Na_2_CO_3_, and the absorbance was measured at 765 nm, after 2 h incubation at room temperature, with a model UV-1601 spectrophotometer (Shimadzu, Milan, Italy). Gallic acid was used as a standard, and the total phenolics were expressed as milligram gallic acid equivalents (GAE)/g extract (DW) ± standard deviation (SD). The mean value of total phenolic content was obtained from triplicate experiments.

### 4.6. Free Radical Scavenging Activity

The free radical scavenging activity of *S. brevibracteata* subsp. *subvelutina* extract was determined using the DPPH (2,2-diphenyl-1-picrylhydrazyl) method [92]. The extract was tested at different concentrations (0.0625–2  mg/mL). An aliquot (0.5  mL) of each sample solution was added to 3  mL of daily prepared methanol DPPH solution (0.1  mM). The optical density change at 517  nm was measured, 20  min after the initial mixing, with a model UV-1601 spectrophotometer (Shimadzu, Milan, Italy). Butylated hydroxytoluene (BHT) was used as a reference compound. The scavenging activity was measured as the decrease in absorbance of the samples vs. the DPPH control solution. The assays were carried out in triplicate, and the results are expressed as the mean radical scavenging activity percentage (%) ± SD and mean 50% inhibitory concentration (IC_50_) ± SD.

### 4.7. Reducing Power Assay

The reducing power of *S. brevibracteata* subsp. *subvelutina* extract was determined according to the method of Oyaizu [93]. The extract was tested in the range of 0.0625–2 mg/mL, and a volume of 1  mL of each sample was mixed with 2.5  mL of phosphate buffer (0.2  M, pH 6.6) and 2.5  mL of 1% K_3_Fe(CN)_6_. The mixture was incubated at 50 °C for 20  min. The resulting solution was cooled rapidly, mixed with 2.5  mL of 10% trichloroacetic acid, and centrifuged at 3000  rpm for 10  min. Finally, the upper layer of the solution (2.5  mL) was mixed with 2.5  mL of distilled water and 0.5  mL of 0.1% FeCl_3_, and the absorbance was measured, after 10  min, at 700  nm; increased absorbance of the reaction mixture indicated increased reducing power. Ascorbic acid and BHT were used as reference standards. The assays were carried out in triplicate, and the results are expressed as ascorbic acid equivalent (ASE/mL) ± SD.

### 4.8. Ferrous Ions (Fe^2+^) Chelating Activity

The Fe^2+^chelating activity of *S. brevibracteata* subsp. *subvelutina* shoot extract was estimated by the method of Decker et al. [94]. The extract was tested at different concentrations (0.0625–2 mg/mL), and ethylenediaminetetraacetic acid (EDTA) was used as the reference standard. Briefly, an aliquot (1 mL) of each sample was added to a solution of 2  mM FeCl_2_ (0.05  mL) and MeOH (0.5 mL). The reaction was initiated by the addition of 5  mM ferrozine (0.1  mL). The mixture was shaken vigorously and left standing at room temperature for 10  min; the absorbance was then measured spectrophotometrically at 562  nm. The control contained FeCl_2_ and ferrozine, complex formation molecules. The assays were carried out in triplicate, and the results are expressed as the mean inhibition of the ferrozine–Fe^2+^complex formation (%) ± SD and IC_50_ ± SD.

### 4.9. Antibacterial Activity

The tests were performed on ten standard and hospital strains of bacteria. Among them, four strains of Gram-positive cocci (*Staphylococcus aureus* ATCC 6538 P, *Staphylococcus epidermidis* S3, *Enterococcus faecalis* ATCC 8040/1, *Enterococcus faecium* 34B/8), one Gram-positive rod strain (*Bacillus subtilis* ATCC 6633), and five strains of Gram-negative rods (*Escherichia coli* ATCC 8739, *Enterobacter aerogenes* 35B, *Enterobacter cloacae* 382/2, *Klebsiella pneumoniae* ATCC 16903, *Pseudomonas aeruginosa* ATCC 27853) were used. The study was carried out using the plate agar dilution method. The tested extract of *S. brevibracteata* subsp. *subvelutina* was dissolved in DMSO at a concentration of 500 mg/mL, from which further dilutions were prepared in CASO Agar, Merck (Darmstadt, Germany), at concentrations ranging from 1.0 to 50.0 mg/mL. Twenty-four-hour bacteria cultures were diluted in CASO broth (Merck). The inoculum, which contained 10^5^ CFU/mL, was applied using a calibrated loop on the surface of the medium with the appropriate concentration of the extract tested and without extract (strain growth control). The incubation was carried out at 37 °C for 24 h. The MIC (minimum inhibitory concentration) was determined as the lowest concentration of extract that inhibits the growth of bacteria on the agar. The MIC value of the reference substance, chloramphenicol (Merck, Darmstadt, Germany), against standard Gram-positive strain *Staphylococcus aureus* ATCC 6538 P was 0.005 mg/mL and against the Gram-negative strain *Escherichia coli* ATCC 8739 was 0.1 mg/mL.

### 4.10. Antifungal Activity

The tests were performed on six standard and hospital fungal strains. Among them, three yeast strains (*Candida albicans* PCM 1409 PZH, *Candida krusei* S220, *Candida quilliermondii* 11), two mould strains (*Aspergillus flavus* 35/1, *Penicillium chrysogenum* ATCC 10106), and *Trichophyton tonsurans* 2, strain belonging to dermatophyte were used. The tested extract of *S. brevibracteata* subsp. *subvelutina* was dissolved in DMSO (500 mg/mL), from which further dilutions were prepared in Sabouraud Agar (Merck, Darmstadt, Germany) at concentrations ranging from 1.0 to 50.0 mg/mL.

Twenty-four-hour yeast cultures were diluted in Sabouraud broth to obtain a cell density of 10^5^ CFU/mL. Mould and dermatophyte cultures were washed off fresh agar slants, and then the suspensions were diluted in the same medium to obtain a 10^5^ CFU/mL density. The cultures of tested strains were inoculated on the agar plates’ surface with the appropriate extract concentration or without extract (strain growth control) using a calibrated loop. Incubation was carried out at 37 °C for 24 h (yeasts) and 5 days of incubation at 25 °C (moulds and dermatophytes). The MIC was determined as the lowest concentration of extract inhibiting the growth of fungal strains on the agar. For amphotericin B, a reference substance (Serva, Heidelberg, Germany), the MIC against the standard strain of *Candida albicans* PCM 1409 PZH was 1.0 mg/mL.

### 4.11. Artemia Salina Leach Lethality Test

Medium lethal concentration (LC_50_) determination was carried out according to the method of Meyer et al. [95]. For the determination, different amounts of *S. brevibracteata* subsp. *subvelutina* extract, opportunely dissolved and then diluted in artificial seawater, were transferred to vials to obtain 10, 100, 500, and 1000 µg/mL final concentrations. Ten brine shrimp larvae (*Artemia salina* Leach), taken 48  h after initiation of hatching in artificial seawater, were transferred to each sample vial, and artificial seawater was added to obtain a final volume of 5  mL. After 24  h of incubation at 25–28 °C, the vials were observed using a magnifying glass, and surviving larvae were counted. The assay was carried out in triplicate for each concentration, and LC_50_ values were determined using Litchfield and Wilcoxon’s method. The toxicity level of the extract was assessed according to the toxicity scale reported by Clarkson et al. [29]. The extract is considered non-toxic if the LC_50_ is more than 1000 µg/mL.

### 4.12. Statistical Analysis

All statistical analyses were performed using the STATISTICA 13.3 software program (TIBCO Software Co., Palo Alto, CA, USA). The differences in biomass growth and metabolite content results between the groups were analysed by ANOVA, followed by a Bonferroni post hoc test and *p*-values < 0.05. The level of significance was established at *p* < 0.05. The results were expressed as means ± SD of the mean.

## 5. Conclusions

For the first time, an in vitro culture of a rare endemic plant species of the genus *Scutellaria, Scutellaria brevibracteata* subsp. *subvelutina,* was established and optimised for good biomass increments and high production of bioactive phenolic compounds. The in vitro biomass obtained by us produces high amounts of specific flavonoids of *Scutellaria* (max. 575 mg/100 g DW) and phenylpropanoid glycosides (max. 457 mg/100 g DW). The high content of these metabolites, known antioxidants, could determine the antioxidant and antimicrobial activity documented by us of the methanolic extract from biomass. In this step of research, we propose an in vitro culture grown in MS medium supplemented with 3.0 mg/L BAP and 1.0 mg/L NAA as a rich source of two groups of antioxidants, flavonoids and phenylpropanoid glycosides. The next step of research should be scaling up in vitro culture in commercially available temporary immersion system bioreactors dedicated to microshoot cultures to propose the biomass for detailed phytochemical analysis and for practical applications in pharmacy, the health food industry, and cosmetology.

## Figures and Tables

**Figure 1 molecules-28-01785-f001:**
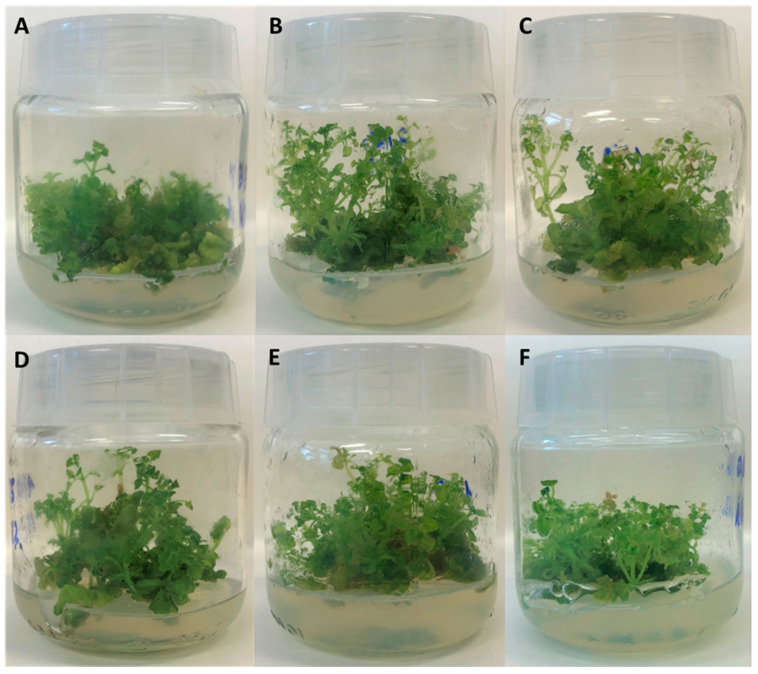
The appearance of *Scutellaria brevibracteata* subsp. *subvelutina* biomasses cultured in vitro depending on the concentration of PGR tested (BAP and NAA) (mg/L) in the MS medium variant (4-week growth cycle). Variant (**A**)—3.0 mg/L BAP and 1.0 mg/L NAA; Variant (**B**)—2.0 mg/L BAP and 2.0 mg/L NAA; Variant (**C**)—1.0 mg/L BAP and 1.0 mg/L NAA; Variant (**D**)—1.0 mg/L BAP and 0.5 mg/L NAA; Variant (**E**)—1.0 mg/L BAP; Variant (**F**)—0.5 mg/L BAP and 2.0 mg/L NAA.

**Figure 2 molecules-28-01785-f002:**
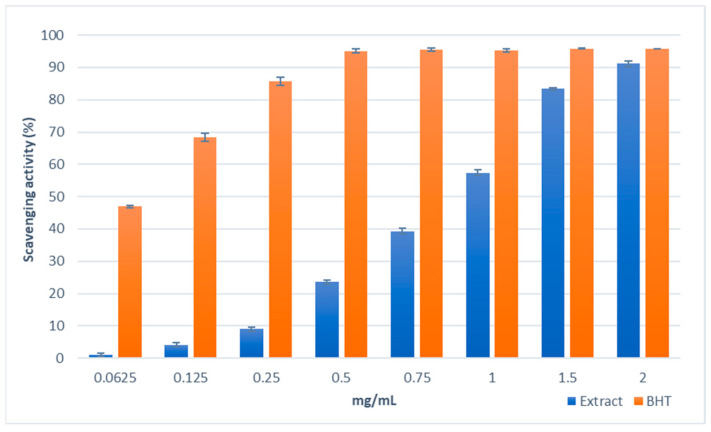
Free radical scavenging activity (DPPH test) of the methanol extract obtained from biomass of in vitro culture of *Scutellaria brevibracteata* subsp. *subvelutina* (3.0 mg/L BAP and 1.0 mg/L NAA). Values are expressed as the mean ± SD (*n* = 3).

**Figure 3 molecules-28-01785-f003:**
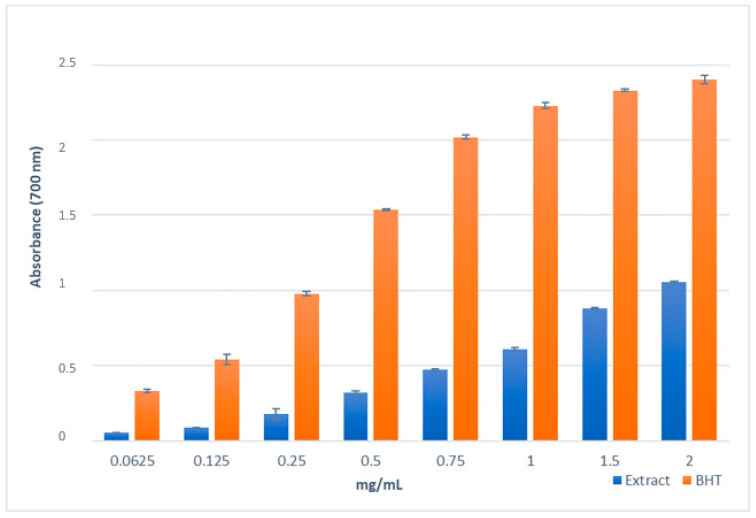
Reducing power of the methanol extract obtained from biomass of in vitro shoot culture of *Scutellaria brevibracteata* subsp. *subvelutina* (3.0 mg/L BAP and 1.0 mg/L NAA). Evaluated by spectrophotometric detection of the Fe^3+^-Fe^2+^ reducing method. Values are expressed as the mean ± SD (*n* = 3).

**Figure 4 molecules-28-01785-f004:**
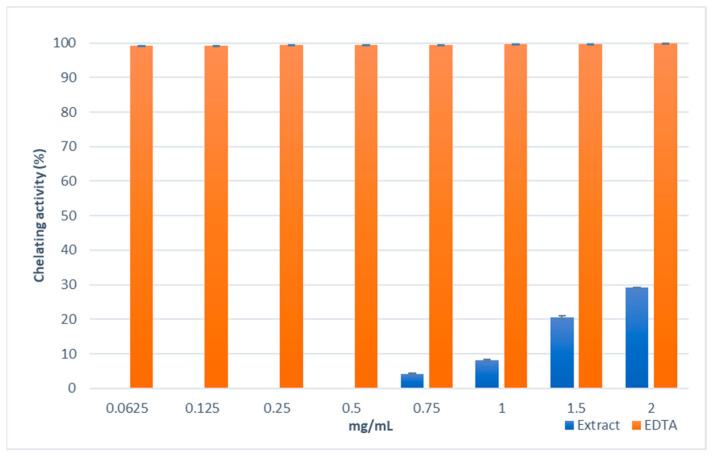
The chelating activity of the methanol extract obtained from the biomass of the in vitro shoot culture of *Scutellaria brevibracteata* subsp. *subvelutina* (3.0 mg/L BAP and 1.0 mg/L NAA). Measured by inhibition of the ferrozine–Fe^2+^ complex formation. Values are expressed as the mean ± SD (*n* = 3).

**Table 1 molecules-28-01785-t001:** Fresh and dry biomass increments of in vitro cultures of *Scutellaria brevibracteata* subsp. *subvelutina* during a 4-week growth cycle (MS medium variants with BAP and NAA).

BAP/NAA (mg/L)	3.0/1.0	2.0/2.0	1.0/1.0	1.0/0.5	1.0/0.0	0.5/2.0
Fresh biomass ^1^ (g)	4.465 ± 0.767	4.616 ± 0.381	4.587 ± 0.570	4.040 ± 0.736	4.923 ± 0.512	4.042 ± 0.318
Dry biomass (g)	0.288 ± 0.051	0.296 ± 0.016	0.287 ± 0.022	0.274 ± 0.044	0.305 ± 0.024	0.275 ± 0.019

^1^ Inoculum 1.2 g.

**Table 2 molecules-28-01785-t002:** Flavonoid content in biomass extracts of *Scutellaria brevibracteata* subsp. *subvelutina* in vitro cultures (MS medium variants with BAP and NAA; 4-week growth cycle).

Flavonoids (mg/g DW)	MS Medium Variant BAP/NAA (mg/L)					
	3.0/1.0	2.0/2.0	1.0/1.0	1.0/0.5	1.0/0.0	0.5/2.0
Baicalein	14.52 ± 3.77	12.63 ± 2.22	13.45 ± 2.21	10.97 ± 1.08	13.42 ± 1.39	12.55 ± 2.79
Baicalin	104.51 ± 14.34 ^abc^	94.57 ± 5.29 ^abcd^	67.89 ± 10.93 ^ade^	70.36 ± 14.23 ^abde^	66.15 ± 11.70 ^abcde^	81.63 ± 9.35 ^acde^
Wogonin	41.58 ± 8.46	32.42 ± 5.80	36.63 ± 4.28	38.27 ± 7.29	38.76 ± 5.38	40.46 ± 12.89
Wogonoside	346.40 ± 53.05 ^ab^	330.44 ± 44.25 ^ab^	194.98 ± 29.78 ^ac^	275.34 ± 61.31 ^abc^	261.06 ± 67.19 ^abc^	273.87 ± 48.25 ^abc^
Scutellarin	35.37 ± 3.76 ^abc^	31.02 ± 5.37 ^ab^	35.67 ± 6.67 ^abc^	34.48 ± 3.87 ^abc^	40.69 ± 2.10 ^abc^	30.99 ± 3.78 ^ac^
Chrysin	32.46 ± 2.27	32.19 ± 3.61	35.97 ± 4.36	38.91 ± 5.80	38.00 ± 6.24	35.50 ± 7.75
Total content	574.84 ± 85.66	533.26 ± 66.54	384.59 ± 58.24	468.33 ± 93.58	458.08 ± 94.00	475.01 ± 84.81

Different letters indicate significant differences (*p* < 0.05).

**Table 3 molecules-28-01785-t003:** Phenolic acid content in biomass extracts of *Scutellaria brevibracteata* subsp. *subvelutina* in vitro cultures (MS medium variants with BAP and NAA; 4-week growth cycle).

Phenolic Acids (mg/g DW)	MS Medium Variant BAP/NAA (mg/L)					
	3.0/1.0	2.0/2.0	1.0/1.0	1.0/0.5	1.0/0.0	0.5/2.0
*p*-Hydroxybenzoic acid	2.09 ± 0.72 ^a^	0.56 ± 0.06 ^b^	0.62 ± 0.07 ^b^	0.79 ± 0.16 ^b^	0.59 ± 0.05 ^a^	1.65 ± 0.26 ^b^
Caffeic acid	2.57 ± 0.39 ^ab^	2.02 ± 0.18 ^abc^	2.20 ± 0.66 ^abc^	2.14 ± 0.21 ^abc^	1.31 ± 0.52 ^ab^	2.43 ± 0.28 ^ac^
Ferulic acid	7.01 ± 3.73 ^abc^	10.14 ± 1.16 ^ab^	6.55 ± 1.57 ^abc^	8.13 ± 0.16 ^abc^	7.93 ± 0.78 ^ab^	5.10 ± 0.39 ^ac^
*m*-Coumaric acid	*	2.76 ± 0.53	*	*	*	2.90 ± 0.55
Total content	11.67 ± 4.84 ^abc^	15.47 ± 1.93 ^bd^	9.37 ± 2.30 ^abc^	11.06 ± 0.53 ^abd^	9.83 ± 1.35 ^abd^	12.09 ± 1.48 ^abc^

*—not detected; different letters indicate significant differences (*p* < 0.05).

**Table 4 molecules-28-01785-t004:** Phenylpropanoid glycosides content in biomass extracts of *Scutellaria brevibracteata* subsp. *subvelutina* in vitro cultures (MS medium variants with BAP and NAA; 4-week growth cycle).

Phenylpropanoid Glycosides (mg/g DW)	MS Medium Variant BAP/NAA (mg/L)					
	3.0/1.0	2.0/2.0	1.0/1.0	1.0/0.5	1.0/0.0	0.5/2.0
Verbascoside	440.55 ± 69.60 ^a^	364.20 ± 36.97 ^ab^	364.31 ± 39.65 ^ab^	362.12 ± 39.96 ^ab^	377.35 ± 45.08 ^ab^	349.73 ± 34.12 ^b^
Isoerbascoside	16.84 ± 3.54	17.19 ± 3.21	19.84 ± 4.91	17.60 ± 4.51	17.90 ± 0.00	15.34 ± 5.17
Total content	457.39 ± 73.13	381.40 ± 40.18	384.14 ± 44.55	379.72 ± 44.47	395.25 ± 45.08	365.07 ± 39.28

Different letters indicate significant differences (*p* < 0.05).

**Table 5 molecules-28-01785-t005:** Antibacterial activity (MIC—minimum inhibitory concentration) of the methanol extract obtained from biomass of in vitro shoot culture of *Scutellaria brevibracteata* subsp. *subvelutina* (3.0 mg/L BAP and 1.0 mg/L NAA)against Gram-positive and Gram-negative bacteria strains.

Bacterial Strains	Species of Microorganisms	Antibiotic Activity (MIC–mg/mL)
Gram-positive bacteria	*Staphylococcus aureus*	20.0
	*Staphylococcus epidermidis*	30.0
	*Enterococcus faecalis*	30.0
	*Enterococcus faecium*	30.0
	*Bacillus subtilis*	10.0
Gram-negative bacteria	*Escherichia coli*	30.0
	*Enterobacter aerogenes*	30.0
	*Enterobacter cloacae*	30.0
	*Klebsiella pneumoniae*	30.0
	*Pseudomonas aeruginosa*	20.0

**Table 6 molecules-28-01785-t006:** Antifungal activity (MIC—minimum inhibitory concentration) of the methanol extract obtained from biomass of in vitro shoot culture of *Scutellaria brevibracteata* subsp. *subvelutina* (3.0 mg/L BAP and 1.0 mg/L NAA).

Fungal Strains	Species of Microorganisms	Antibiotic Activity (MIC—mg/mL)
Yeast-like fungi	*Candida albicans*	30.0
	*Candida krusei*	30.0
	*Candida quilliermondii*	30.0
Moulds	*Aspergillus flavus*	40.0
	*Penicillium chrysogenum*	30.0
Dermatophytes	*Trichophyton tonsurans*	7.5

**Table 7 molecules-28-01785-t007:** PGR composition of the Murashige and Skoog medium variants applied in the experiment with *Scutellaria brevibracteata* subsp. *subvelutina* in vitro cultures.

MS Medium Variant		A	B	C	D	E	F
PGR (mg/L)	BAP	3.0	2.0	1.0	1.0	1.0	0.5
	NAA	1.0	2.0	1.0	0.5	0.0	2.0

## Data Availability

Not applicable.

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
