# Peer review of "In Vitro Cultures of Scutellaria brevibracteata subsp. subvelutina as a Source of Bioactive Phenolic Metabolites"

_molecules, 2023, doi:10.3390/molecules28041785_

Round 1

Reviewer 1 Report

The current submission reported in vitro cultures of Scutellaria brevibracteata subsp. subvelutina as a source of bioactive phenolic metabolites. Results show that the in vitro biomass obtained by us produces high amounts of specific flavonoids of Scutellaria (max. 575 mg/100g DW) and phenylpropanoid glycosides (max. 457 mg/100g DW).  Meanwhile, MS medium supplemented with 3.0 mg/L BAP and 1.0 mg/L NAA as a rich source of two groups of antioxidants, flavonoids and phenylpropanoid glycosides was proposed.  The extract showed good antioxidant activity as well as anti-microbial activity. The submission can be accepted after minor revision.

1. Table 3 and 4: why the ratio of BAP/NAA affected phenolic acids & blycosides production should be discussed at molecular biological level.

2. Statistical analysis should be applied for Table 4.

3. Why the authors evluated DPPH, reducing power as well as chelating activity should be explain. Please provide a good discussion if you are targeting different compound within the extract.

4. Table 5: Seems like there are difference of the Antibacterial activity toward gram positive and negative strains, can you explain why? The possible mechanism?

5. line 249: The data should be revealed even in the supplementary.

Reviewer 2 Report

This manuscript is about the production of flavonoids by Scutellaria shoot cultures, and I have few considerations:

- Please, review all text to put all scientific names in italic.

- What were the criteria to choose the values of BAP and NAA in Table 7? 

Why the increase of BAP from 1 to 3 mg/L also better yield on flavonoids? What metabolic way was activated or inhibited? Authors could improve the explanation
